# State-level variation of initial COVID-19 dynamics in the United States

**Easton R. White** [1,2]*, **Laurent Hébert-Dufresne** [3,4]

**1** Department of Biology, University of Vermont, Burlington, VT, United States of America, **2** Gund Institute for Environment, University of Vermont, Burlington, VT, United States of America, **3** Department of Computer Science, University of Vermont, Burlington, VT, United States of America, **4** Vermont Complex Systems Center, University of Vermont, Burlington, VT, United States of America

* Easton.White@uvm.edu

## Abstract

During an epidemic, metrics such as $R_0$, doubling time, and case fatality rates are important in understanding and predicting the course of an epidemic. However, if collected over country or regional scales, these metrics hide important smaller-scale, local dynamics. We examine how commonly used epidemiological metrics differ for each individual state within the United States during the initial COVID-19 outbreak. We found that the detected case number and trajectory of early detected cases differ considerably between states. We then test for correlations with testing protocols, interventions and population characteristics. We find that epidemic dynamics were most strongly associated with non-pharmaceutical government actions during the early phase of the epidemic. In particular, early social distancing restrictions, particularly on restaurant operations, was correlated with increased doubling times. Interestingly, we also found that states with little tolerance for deviance from enforced rules saw faster early epidemic growth. Together with other correlates such as population density, our results highlight the different factors involved in the heterogeneity in the early spread of COVID-19 throughout the United States. Although individual states are clearly not independent, they can serve as small, natural experiments in how different demographic patterns and government responses can impact the course of an epidemic.

## Introduction

The global COVID-19 outbreak caused by the SARS-CoV-2 virus began in Wuhan, China in late 2019 [1]. As of Sep 19[th], over 30 million cases and nearly one million deaths have been reported across the globe. There have been several sets of efforts to track the progression of the outbreak across the world and within countries. Early in the outbreak, the Johns Hopkins University Center for Systems Science and Engineering (CSSE) started compiling data from various sources, including the US Center for Disease Control and the World Health Organization, to present a global picture of COVID-19 cases and deaths [2]. Later in the outbreak, large distributed teams were necessary to compile and curate epidemiological case data at the individual level [3]. These efforts have allowed for international scientific research and political decision-making. Although data are collected at local scales (e.g. within hospitals), in an

**Data Availability Statement:** All code and corresponding data is freely available at \url{https://github.com/eastonwhite/COVID19_US_States}. The original raw data has been compiled by the Johns Hopkins University Center for Systems

Science and Engineering at (\url{https://github.com/CSSEGISandData/COVID-19}).

**Funding:** L.H.-D. acknowledges support from the National Institutes of Health 1P20 GM125498-01 Centers of Biomedical Research Excellence Award. The funders had no role in study design, data collection and analysis, decision to publish, or preparation of the manuscript.

**Competing interests:** The authors have declared that no competing interests exist.

emerging pandemic, data are typically reported at the country or regional level. This allows for interesting comparisons between countries [4–6] and for information from an earlier affected country to be used to slow the outbreak in other places. For instance, South Korea was able to "flatten their outbreak curve" through early and widespread testing as well as strict quarantine policies [7]. However, country-level analyses still hide more local dynamics that are important to the overall epidemic progression [8, 9]. For example Lin et al. (2020) found that, in China, traffic control and social distancing measures did not work effectively everywhere. Instead, these measures depended on income and population size [9].

Spatial heterogeneity is important for population dynamics generally [10–12] and in particular for understanding the progression of infectious disease dynamics [13, 14]. Spatial heterogeneity can include differences in local population density, movement patterns, suitability of environmental conditions for transmission, among other factors [13–15]. These heterogeneities can impact the spread of any infectious disease. For example, Keeling et al. [16] showed how spatial distribution and size of farms affected the 2001 UK Foot and Mouth Epidemic, Apolloni et al. studied the role of age structure and travel patterns for 2009 H1N1 pandemic, and the GLEAM framework has been used to model diseases from Zika to COVID-19 while including population density and mobility data [17–19].

Here we provide a descriptive analysis of the reported progression of COVID-19 at the state level within the United States. We examine how commonly-used metrics, focusing on doubling time, can vary by state. Clearly, controlled and randomized experiments of COVID-19 spread are not possible. Therefore, although states are not independent units, we can use state-level data to understand the progression of the outbreak as detected by heterogeneous testing systems across different replicates within a country [20]. There are three periods of interest during an epidemic: the initial exponential phase, the slowing of the curve before before the peak, and the decrease in new cases after the peak. We expect all three of these phases to look different for each state depending on both demographics and interventions. Specifically, we hypothesize that the doubling time of the outbreak in each state will be correlated with population density and metrics of overall population health. In the middle of the outbreak, after government interventions have been enacted, we hypothesize that doubling times will be correlated with a combination of demographics and these government interventions. We also assess two correlates not traditionally tied to work on infectious disease dynamics: volunteerism rates and community tightness. We examined volunteerism rates as a measure of how willing individuals may be willing to adjust their behavior to help reduce overall disease spread. Community tightness is a measure indicating adherence to rules and strong cultural norms [21]. Recently, Gelfand et al. [22], found that countries with both efficient governments and high tightness scores were more effective in reducing the spread of COVID-19 and reducing mortality rates. In this paper, we focus on the early dynamics of the pandemic, where inadequate and heterogeneous testing capacities are as, if not more, important in driving epidemic dynamics than geospatial differences [23]. This allows us to test for a wide-range of potential correlates—testing, intervention, population density, behavior, etc.—all potentially driving early reported data. That being said, some of the results highlighted here can also be useful in understanding the varied decay patterns of COVID-19 [24] as well as its resurgence in different places [25]. Indeed, testing improved in all states but human behavior and control policies remain heterogeneous and inconsistent [26].

## Materials and methods

We used data on the daily number of cases compiled by Johns Hopkins University Center for Systems Science and Engineering [2]. The United States experienced an initial exponential

growth in the number of cases, especially since February 29th (S1 Fig) and slowed starting in mid-March before rising again in summer. In addition, outbreaks have been reported as schools, including college campuses, have reopened [2]. Country-level results, however, hide underlying dynamics within each state [8, 9].

Therefore, we examined how the number of cases changed over time within each state. To properly compare the progression of the epidemic across states, we looked at the log number of cases since the first day a state reported 25 (after which exponential curves were more reliable) cases (Fig 2). Although no starting value is perfect, we chose 25 cases as the modeling results were qualitatively more similar after 25 cases. In other words, models with a starting value of less than 25 cases were not very consistent, likely due to early differences in reporting and case definition [27]. On a log scale, a straight line of the cases over time indicates exponential growth where the slope of the line is the exponential growth parameter. This slope informs us about the growth of the epidemic, and is related to a useful quantity known as doubling time: the expected length of time required to see the number of cases double. A longer doubling time in reported cases is therefore preferred as it suggests a slower spread of the disease.

We calculated the doubling times throughout the course of the epidemic within each state. We then looked at the early (first 7 days since 25 cases) and overall doubling times (first 21 days since 25 cases) as our two response variables. We did not examine the doubling time after 21 days as some states were already recorded a decrease in the daily number of cases per day at that point (S2 Fig). The early doubling time should not be strongly correlated with government interventions as these were often implemented late and there is a delay in which effects from government actions are detectable [28].

We used a simple linear regression to evaluate possible predictors of early and overall doubling times. We evaluated all possible combinations of explanatory variables using stepwise model selection. We compared models using Akaike Information Criterion (AIC) to determine the best fitting model while accounting for model complexity. We did not examine interactions between predictor variables given the limited number of data points. All analyses were performed in R [29]. To check model assumptions, we examined the Pearson residuals plotted against model predictions and time (S4 Fig). In line with past work, we also did not include variables that were strongly correlated (greater than 0.7) in the same model [30, 31]. Specifically, several demographic variables (e.g. population percent in rural areas versus population percent in urban areas) were strongly correlated. In those cases, we arbitrarily only included one variable in the model.

## Demographic correlates

Following Chin et al. [8], we collected demographic, health, education, and income variables from a variety of sources:

- population density and percent in rural areas (2010 US Census Bureau https://www.census.gov/programs-surveys/geography/guidance/geo-areas/urban-rural/2010-urban-rural.html)

- percent of population over age 65 (U.S. Census Bureau, Vintage 2018 Population Estimates https://www.prb.org/which-us-states-are-the-oldest/)

- life expectancy, income per capita, and expected years of schooling (Global Data Lab https://globaldatalab.org/shdi/download/2018/indicators/USA/?interpolation=0&extrapolation=0&nearest_real=0&format=csv)

- yearly flu vaccination rate (ChildVaxView CDC https://worldpopulationreview.com/states/vaccination-rates-by-state/)

- volunteer rate as a measure of willingness buy-in to restrictive policies that might benefit the community (2015 Corporation for National and Community Service data https://www.nationalservice.gov/vcla/state-rankings-volunteer-rate)

- "tightness" score, where a high tightness indicates a state with "many strongly enforced rules and little tolerance for deviance" [21]

- testing rates by state (COVID Tracking Project https://covidtracking.com/)

### State government interventions

Much of the response to COVID-19 in the United States has been done at the state, as opposed to federal, government level [20, 32]. Several policies started at city or county levels before being implemented across an entire state [33, 34]. Although interventions, guidelines and policies have varied, some slowly became almost ubiquitous across all states. We focused on five such state-wide mandates: declaring a state of emergency, limiting gatherings (usually to 10 people or less), closing schools, restricting restaurants, restricting businesses generally, and stay-at-home mandates. This is also the general sequence in which state-wide mandates were implemented (see Fig 1). To quantify the timing of the intervention relative to COVID-19, we used information collected by Adolph et al. [20] on whether or not a state had mandated a specific action by the first day they had 25 or more cases. We adjusted this number to 150 cases for more severe restrictions like closing all non-essential business or stay at home mandates). We follow Adolph et al. [20] and use the first date of gathering restrictions announced, regardless of the size of the gatherings restricted.

### Results

We found considerable differences between states in how the outbreak initially progressed (Fig 2). These doubling times are, of course, changing over time. We found that doubling times for all states did increase with time and that the heterogeneity between states was reduced (see the change in spread of lines between Fig 2a and 2b). We mapped doubling time across the US and found regional differences where the West and Northeast have seen large doubling times, i.e. slower outbreak dynamics (S3 Fig).

### Early dynamics

Given the large heterogeneity between states early on (Fig 2), we examined correlates of doubling time for only the first seven days after a state reached 25 total cases. Comparing the doubling times across states, we found that lower population density along with higher flu vaccination rates and wealth were associated with slower epidemics. (Fig 3, Table 1).

### Overall dynamics

We then examined the overall (first 21 days since 25 total cases) doubling times at the state level. Except for population density and percent of population living in rural areas, we found that demography, education, and wealth were poor predictors of the state-level overall doubling times (Fig 3, Table 1). Therefore, we also examined the correlation between doubling time and state government interventions. Of state-wide mandates, only restricting restaurant operations was a significant (at $\alpha < 0.05$) predictor of doubling time (Fig 4, Table 1). However, these restrictions were also additive, as states that implemented more actions early had higher doubling times (Fig 4f). The ordering of these restrictions was also fairly consistent between

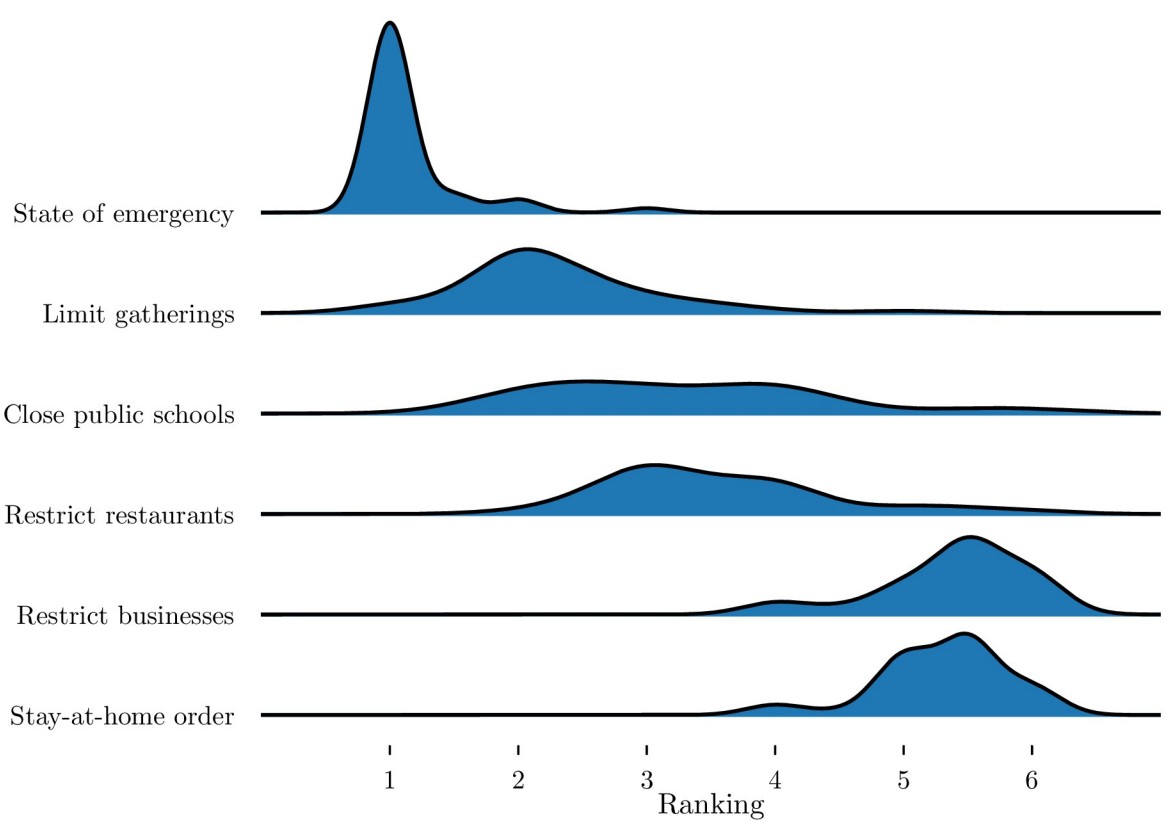

**Fig 1. Rank distribution of different interventions.** Per state, every intervention is given a rank from 1 to 6 depending on when it was implemented (1 being the first put into place) and ties are given an average rank (e.g. 2.5 for tied 2nd and 3rd rank).

states (Fig 1). While declaring a state of emergency is an obvious first intervention, closing public schools tend to be implemented at different times across states. More importantly, by the time local governments restrict businesses and declare a stay-at-home order, all other interventions tend to have already been implemented (Fig 1, [20]). Lastly, a state's tightness score, a measure indicating adherence to rules and strong cultural norms [21], was also negatively correlated with doubling time with tighter states having lower doubling times (Fig 4, Table 1). The overall doubling time was not strongly correlated with overall tests rates for each state (Table 1, Fig 5).

## Discussion

A lot of work on the COVID-19 pandemic has drawn on comparisons between countries to evaluate the effectiveness of different interventions [6, 35, 36]. However, this hides underlying variation within each country. We investigated the variation in how the epidemic unfolded across states in the United States. We found substantial variation between states, particularly in the early phase of the epidemic (Fig 2). We found that the early dynamics (first seven days) were most strongly correlated with population density, percent of population living in rural areas, income, and yearly flu vaccination rate (Fig 3). This is in line with other work on COVID-19 that found population demography important at the US county level [37] and in other countries. Interestingly, while the correlation between the intensity of a contagious respiratory infection and population density is intuitive, our findings are actually contrary to patterns observed during Influenza seasons [38] and for COVID-19 in China [39]. The different

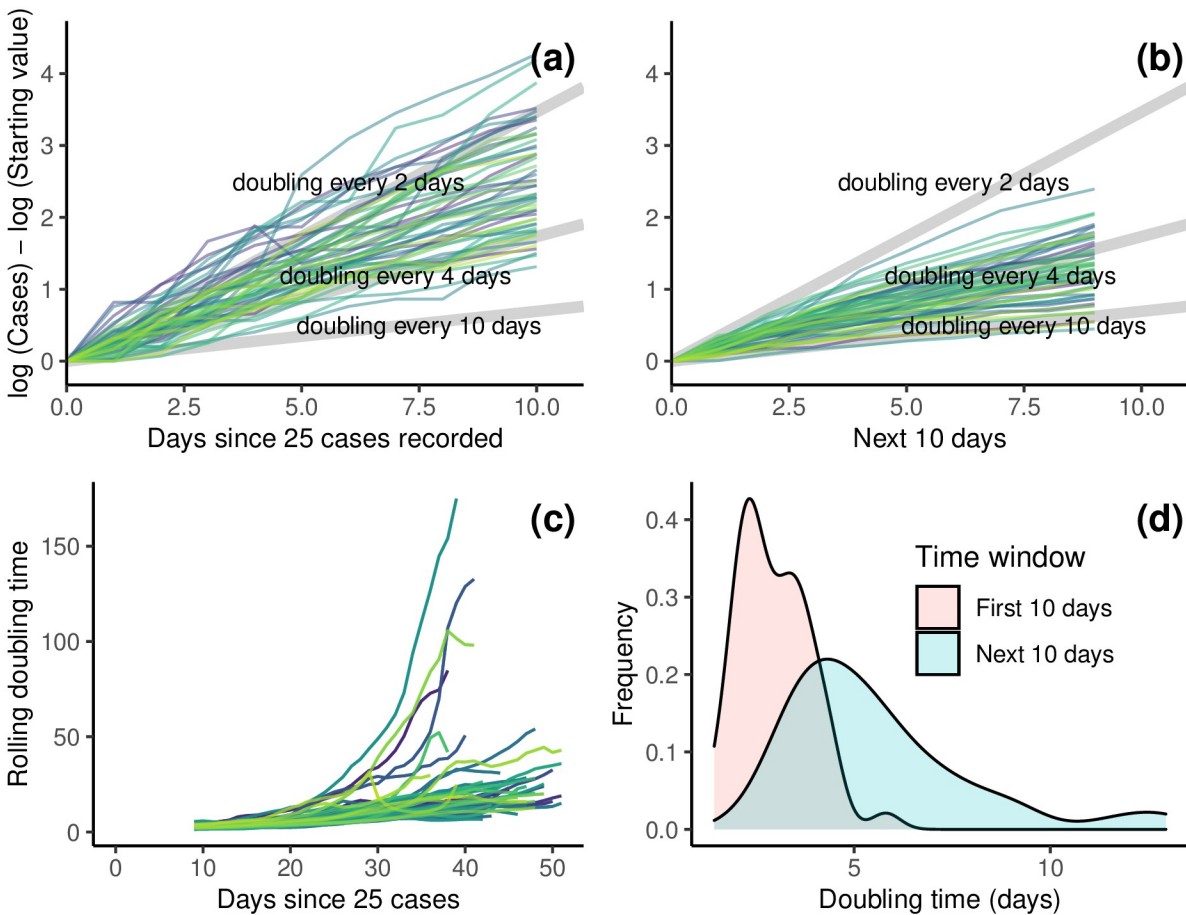

**Fig 2.** (a) The log number of cases over time for each individual state for the 10 days since their first day of 25 total cases. (b) The log number of cases over time for each individual state for the next 10 days. The light grey diagonal lines represent the growth trajectory for doubling times of 2, 4, and 10 days. The log number of the starting value (initial number of cases on first day when at least 25 cases were recorded) had to be subtracted on the y-axis to standardize the graph across states. (c) Rolling doubling times calculated over 10-day windows for each individual states. (d) Distributions of state-level doubling times early and more recent in the course of the outbreak.

behavior observed here could be due to COVID-19 being less reliant on environmental factors and mostly driven by social behaviours. Similarly, it also suggests that patterns observed in China might be driven by the timing and scale of government interventions and not by density itself given the opposite behavior observed in the United States.

In addition to population density, the early doubling time was also correlated with flu vaccination rate and average income ("GNI per capita" in Table 1). For each percentage increase in vaccination rate, there was a 0.11 day difference in doubling time. For example, a states with at least 0.85 vaccination coverage had an average early doubling time of 3.5 days compared to 2.5 days for states with vaccination rates under 0.85. We hypothesize this has to do with general health and access to healthcare for people in more vaccinated states. In addition, the early doubling time increased with average income per capita (Table 1). We would also hypothesize that average income also related to access to healthcare and baseline health. In addition, other work has shown that those from higher income brackets were able to socially distance earlier and to a larger degree [25].

The overall doubling time (first 21 days of the epidemic) in each state was also correlated with demographic parameters. However, in line with past work [28, 40, 41], we show early

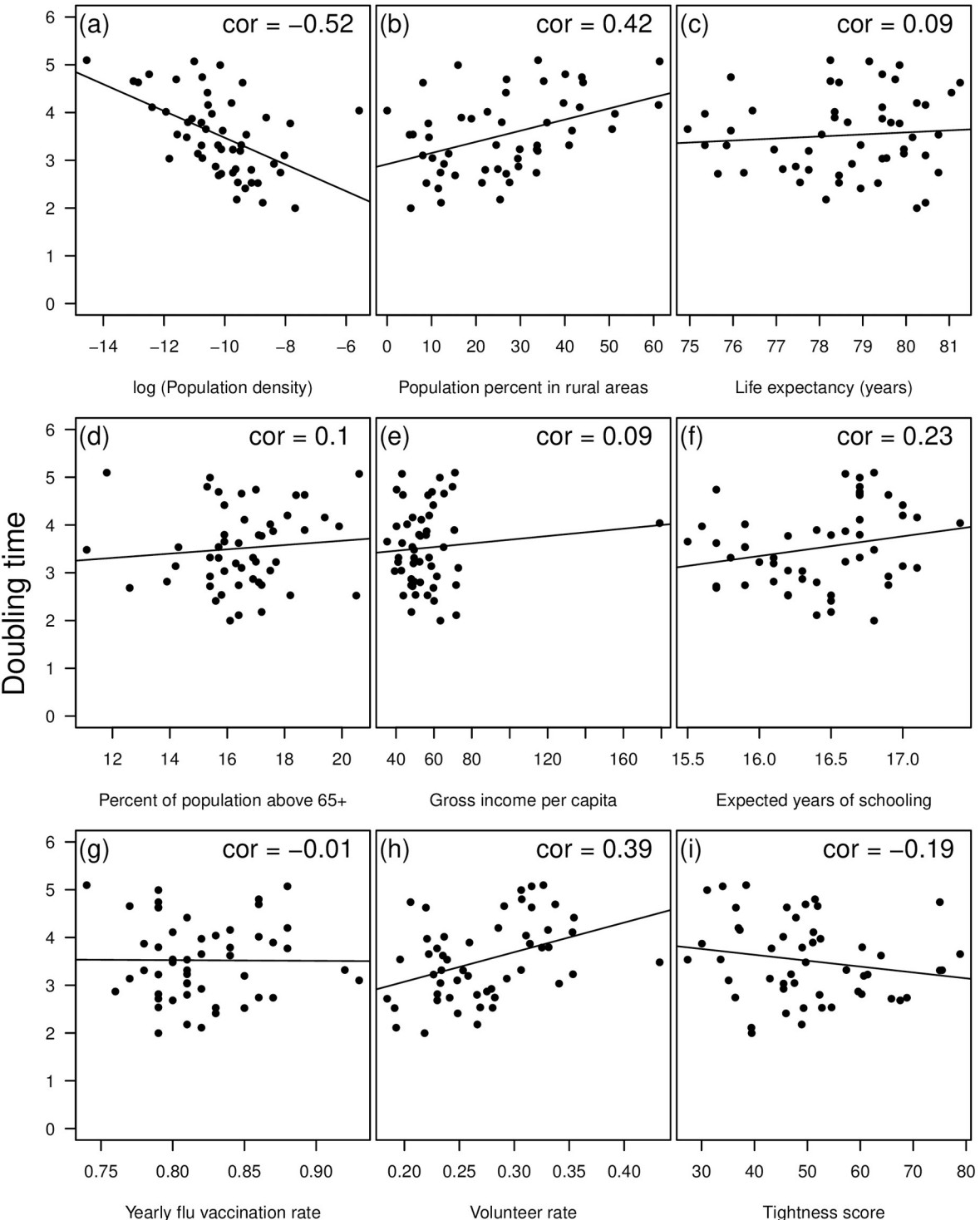

**Fig 3.** Doubling time (in number of days) versus (a) log (population density), (b) population density in rural areas, (c) life expectancy (years), (d) percent of population above age 65, (e) gross income per capita (in 1000s USD), (f) expected years of schooling, (g) yearly flu vaccination rate, (h) volunteer rate, and (i) tightness score.

**Table 1. Best fitting linear models (according to AIC) and corresponding parameter estimates for the doubling time both early (first 7 days since 25 cases) and for the first three weeks.** A parameter was not included in the table if it was not selected in the best fitting linear model. The overall model included the following parameters (see Methods for more detail): log(Population density), population percent in rural areas, percent of population over age 65, influenza vaccination rate, gross income per capita, expected number of years of schooling, community tightness score, tests per capita (at the end of 7 days or 21 days, respectively), and binary variables for the presence or absence of each social distancing restriction (see Fig 4).

| | *Dependent variable*: | |
| --- | --- | --- |
| | **Early doubling time** | **Overall doubling time** |
| Restrict restaurants | | 0.560** |
| | | (0.150, 0.969) |
| log (Population density) | −0.342*** | −0.293*** |
| | (−0.467, −0.218) | (−0.421, −0.166) |
| Vaccination rate | 11.308*** | |
| | (6.914, 15.702) | |
| GNI per capita | 0.037*** | |
| | (0.021, 0.054) | |
| Population percent in rural areas | | 0.014* |
| | | (−0.0004, 0.028) |
| Tightness score | | −0.018** |
| | | (−0.032, −0.004) |
| Constant | −12.041*** | 0.901 |
| | (−16.384, −7.697) | (−0.521, 2.324) |
| Observations | 50 | 50 |
| $R^2$ | 0.554 | 0.589 |
| Adjusted $R^2$ | 0.525 | 0.553 |

Note:

*$p < 0.1$;

**$p < 0.05$;

***$p < 0.01$

doubling time was also associated with government actions, most notably restrictions on restaurant operations (Fig 4, Table 1). For example, a parameter estimate of 0.560 for restaurant closures implies that states with restaurant closures experienced a half-day increase in their doubling time. The specific timing of restrictions was not as important as simply a mandate to limit operations early in the epidemic (before 25 cases were recorded). This does not provide causal evidence for government actions as a key factor in controlling the disease spread, only an association. Instead, business restrictions and school closures could simply be coarse metrics of actual social distancing.

Lastly, after accounting for other demographic variables, a state's "tightness" score was also correlated with overall doubling time (Table 1). A state with a high tightness score has "many strongly enforced rules and little tolerance for deviance" [21]. Recently, Gelfand et al. [22], showed that countries with efficient governments and high tightness scores in reducing the spread of COVID-19 and its mortality rate. Thus, we expected that states with highly enforced rules should have higher doubling times compared to "loose" states. Instead, we found the opposite where tight states had low doubling times and consequently faster disease spread. We hypothesize this may be the result of people in tight cultures finding it more difficult to adjust their behavior when new rules are imposed. This could lead individuals to protest or avoid closures. Althouse et al. (2020) showed that states with higher tightness scores had decreased

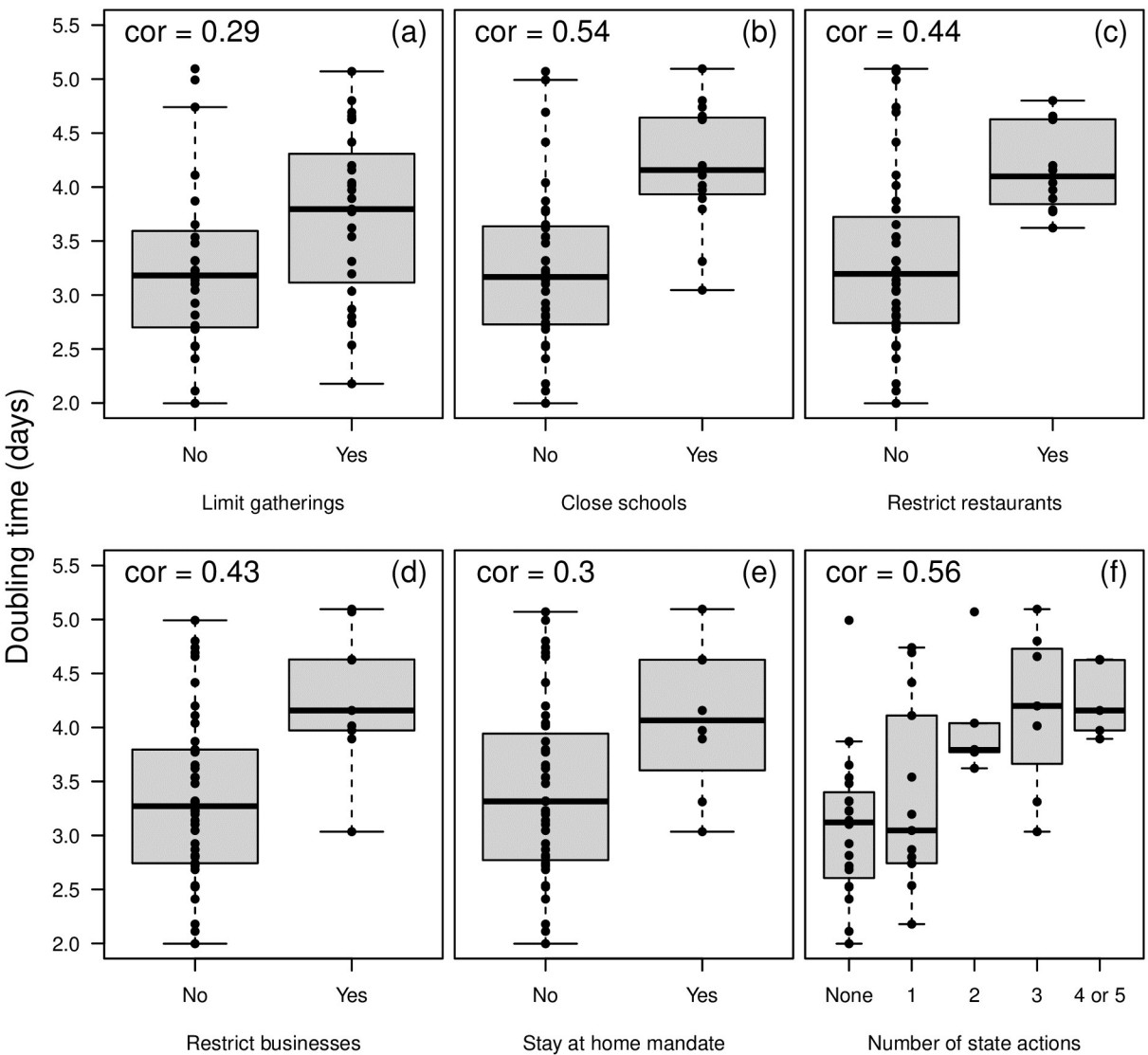

**Fig 4.** Doubling time (in number of days) across the US states for five different statewide government restrictions: (a) limit gatherings (usually to less than 10 people, but see Methods) by first day of 25 cases, (b) close public schools by first day of 25 cases, (c) restrict restaurants by first day of 25 cases, (d) restrict non-essential businesses by first day of 150 cases, (e) stay at home order by first day of 150 cases, and (f) total number of restrictions before number of cases threshold.

number of visits to non-essential business, but also increased distance traveled for any visits that did take place. Using a simple mathematical model, they were able to show that this type of behavior leads people to cluster and can cause outbreaks to occur [26]. Future work should focus on better understanding this relationship through mobility data and more detailed mechanistic modeling.

There are several limitations to this work that require further investigation. In many places, government restrictions went into effect at a smaller scale, often in cities or counties, before implemented at the state level [34]. Thus, county-level analyses are a natural extension of this work [33, 37]. In addition, government restrictions are a crude measure that does not capture actual social distancing. For instance, work has shown that individuals reduced their movement often before certain government actions were implemented. Similarly, public buy-in to

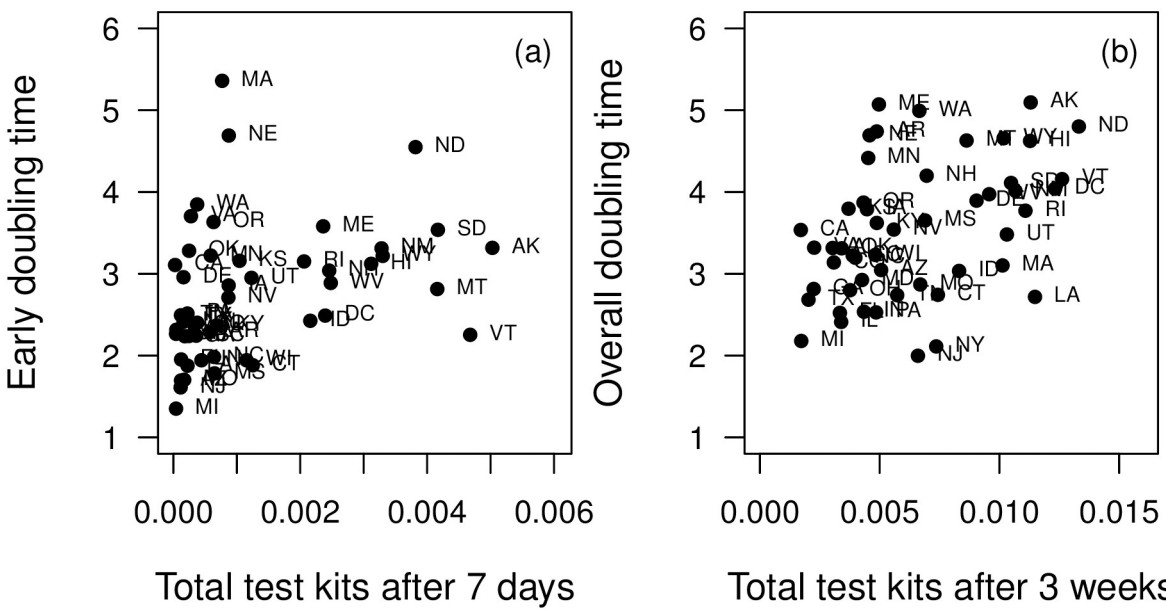

**Fig 5.** Doubling time (in number of days) for each state according to the total tests conducted per capita (a) early in the outbreak (within the first week since 25 cases) and (b) for within the first three weeks.

various restrictions has varied across the US [42]. Testing rates were not a significant predictor of doubling time (Table 1). However, each state implemented their own policies for who could receive a test and these policies changed throughout the course of the epidemic [43]. Thus, this heterogeneity in testing may influence doubling time (Fig 5) and should be evaluated further [43].

## Conclusion

We found a large degree of heterogeneity in the reported number of COVID-19 cases over time across the United States. After state-level government actions were implemented, doubling time was most strongly correlated to social distancing restrictions, in particular restaurant operations (Fig 4). More detailed work will be needed to understand how these dynamics differ within each state, especially as many government actions started on more local scales [33, 34]. Similar to the implementation of restrictions, states have loosened, and sometimes reimposed, restrictions, but this varies across the country. This will present another natural experiment on the effect of government actions on the course of the epidemic.

## Supporting information

**S1 Fig. COVID-19 cases over time for the US as a whole.**
(TIF)

**S2 Fig. COVID-19 cases over time for each US state.**
(TIF)

**S3 Fig. Map of COVID-19 case doubling time (first three weeks since 25 cases) for each US state.**
(TIF)

**S4 Fig. Residual diagnostic plots for the best fitting models for both the early (first 7 days) and overall (first 21 days) doubling times.**
(TIFF)

## Acknowledgments

We thank three anonymous reviewers for feedback on an earlier version of the manuscript.

## Author Contributions

**Conceptualization:** Easton R. White, Laurent Hébert-Dufresne.

**Data curation:** Easton R. White.

**Investigation:** Easton R. White, Laurent Hébert-Dufresne.

**Methodology:** Easton R. White, Laurent Hébert-Dufresne.

**Validation:** Easton R. White.

**Visualization:** Easton R. White, Laurent Hébert-Dufresne.

**Writing – original draft:** Easton R. White, Laurent Hébert-Dufresne.

**Writing – review & editing:** Easton R. White, Laurent Hébert-Dufresne.

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
