## [Decision Letter · Decision Letter 0]

13 Jul 2020

PONE-D-20-13138

State-level variation of initial COVID-19 dynamics in the United States

PLOS ONE

Dear Dr. White,

Thank you for submitting your manuscript to PLOS ONE. After careful consideration, we have decided that your manuscript does not meet our criteria for publication and must therefore be rejected.

Specifically: the reviewer notes a number of methodological flaws that suggest the findings are biased.

I am sorry that we cannot be more positive on this occasion, but hope that you appreciate the reasons for this decision.

Yours sincerely,

Jeffrey Shaman

Academic Editor

PLOS ONE

Reviewers' comments:

Reviewer's Responses to Questions

**Comments to the Author**

1. Is the manuscript technically sound, and do the data support the conclusions?

Reviewer #1: No

2. Has the statistical analysis been performed appropriately and rigorously? 

Reviewer #1: No

3. Have the authors made all data underlying the findings in their manuscript fully available?

Reviewer #1: Yes

4. Is the manuscript presented in an intelligible fashion and written in standard English?

Reviewer #1: Yes

5. Review Comments to the Author

Reviewer #1: In this paper, the authors explore differences in state-level COVID outcomes and policy responses. There have already been a number of papers exploring exactly the same topic (e.g. Courtemanche et al. 2020) and with much better modeling. I know PLOS ONE has a policy that novelty is not a criterion, but what these other papers highlight is that the models in this paper are very likely misspecified.

The authors note the independent variables are highly correlated with one another and only remove them if they are correlated at 0.7 or more. This is still very high collinearity so I'm not sure we can trust the model results. The fact that the authors' note that their results are contrary to those for influenza and for analyses of COVID in other countries, makes me worry that the model is misspecified.

The policy variable is very crude, equal to 1 if policy interventions happened before 25 cases and 0 if they happened after. Why not use the number of days prior to or before some arbitrary threshold? That's the way the IHME model works (https://covid19.healthdata.org).

I'd also like the authors to justify the arbitrary threshold of 25 cases. And since population varies so much across states, why use number of cases rather than percent of population?

The models very likely suffer from omitted variable bias. A better analysis would forego the state level independent variables and instead use variation over time and fixed effects to account for differences between states.

6. PLOS authors have the option to publish the peer review history of their article (what does this mean?). If published, this will include your full peer review and any attached files.

Reviewer #1: No

- - - - -

---

## [Author Response · Author response to Decision Letter 0]

20 Jul 2020

We have included a revised manuscript and detailed response to reviewer concerns.

---

## [Decision Letter · Decision Letter 1]

21 Aug 2020

PONE-D-20-13138R1

State-level variation of initial COVID-19 dynamics in the United States

PLOS ONE

Dear Dr. White,

Thank you for re-submitting your manuscript to PLOS ONE. We have obtained comments from two reviewers and  feel, based on their reviews, that the manuscript has merit but does yet not fully meet PLOS ONE’s publication criteria as it currently stands. Therefore, we invite you to submit a revised version of the manuscript that addresses the points raised during the review process.

Please respond to the reviewer comments on a point-by-point basis and revise the manuscript accordingly.

We look forward to receiving your revised manuscript.

Kind regards,

Jeffrey Shaman

Academic Editor

PLOS ONE

Journal Requirements:

2) We note that your manuscript is not formatted using one of PLOS ONE’s accepted file types. Please reattach your manuscript as one of the following file types: .doc, .docx, .rtf, or .tex (accompanied by a .pdf).

If your submission was prepared in LaTex, please submit your manuscript file in PDF format and attach your .tex file as “other.”

3) We note that Figure S3 in your submission contains map images which may be copyrighted.

We require you to either (a) present written permission from the copyright holder to publish this figure specifically under the CC BY 4.0 license, or (b) remove the figure from your submission:

a. You may seek permission from the original copyright holder of Figure S3 to publish the content specifically under the CC BY 4.0 license.

Reviewers' comments:

Reviewer's Responses to Questions

**Comments to the Author**

1. If the authors have adequately addressed your comments raised in a previous round of review and you feel that this manuscript is now acceptable for publication, you may indicate that here to bypass the “Comments to the Author” section, enter your conflict of interest statement in the “Confidential to Editor” section, and submit your "Accept" recommendation.

Reviewer #2: (No Response)

Reviewer #3: (No Response)

2. Is the manuscript technically sound, and do the data support the conclusions?

Reviewer #2: Partly

Reviewer #3: Yes

3. Has the statistical analysis been performed appropriately and rigorously? 

Reviewer #2: I Don't Know

Reviewer #3: Yes

4. Have the authors made all data underlying the findings in their manuscript fully available?

Reviewer #2: Yes

Reviewer #3: Yes

5. Is the manuscript presented in an intelligible fashion and written in standard English?

Reviewer #2: Yes

Reviewer #3: Yes

6. Review Comments to the Author

Reviewer #2: While there seem to be interesting and overall likely well done analyses, there are places where the methods are underspecified to the point where it is unclear if the methods are adequate.

Major points

• While I understand the need for stepwise regression and concerns about correlation. It is unclear how the authors selected variables when there were highly correlated variables. If two variables are highly correlated, it would be likely (although not absolutely true) that they would either both meet stepwise inclusion or neither.

• Please clearly list all candidate predictors (state variables as well as policies) that were considered as part of the stepwise regression.

• Please provide more context for the inclusion of the volunteer rate in the methods. This inclusion seems interesting but the reader doesn’t have context for this variable. Similarly, the tightness score (this comes through a little more in the discussion, but explaining in methods would be helpful)

• Please give more information regarding the gathering limitations. You say “usually to 10 people” but the reality is that almost all states started at 250 or 100, then decreased to 50, then 10, some went to smaller numbers too. So what date are you using? The first date or the date the states went to 10 people?

• The references (and at times the text) need to updated. I believe publishing on the beginning of the pandemic is useful but putting these findings into context of other publications around the same time period is useful. A prior reviewer mentioned the Courtemanche manuscript, there is also the Hsiang manuscript in Nature. This submitted manuscript confirms many of the findings of Auger re schools in JAMA. Thus, I think the concerns raised by prior reviewers that the model is not correctly specified are not founded.

In terms of updating the manuscript, the second sentence discusses the case counts as of April 29th. The reader will be confused by this statement given how different we are currently. I see why you are presenting this data as this is at the end of the study period but you need to first introduce the study period and then give context of the numbers.

Similarly, you need to at least acknowledge the latest phase of the pandemic (lines 40-45) there is another rise in cases/deaths after reopening. I understand that you are not focused on that time period but it is strange to not acknowledge it. In fact you state the numbers have “slowed since mid-March” (line 50) which is simply not true.

• Results—Table 1line 123. You state that demography, education, etc are poor predictors and cite table 1; yet none of these variables are in table1.

• Please put in context the parameter estimates from Table 1 into the text. What does a doubling time parameter estimate of 0.492 for schools mean??? This would be helpful for the entire table.

• It is kind of a big deal that testing wasn’t accounted and is only listed as a limitation. The statement “overall doubling time was not strongly correlated with overall tests rates for each state” line 135 is woefully underspecified. (the text says figure 4, but It appears to refer to figure 5). The reader has no idea what is meant by test kits per capita. Is this the number of tests per people over some time? What time period? Why not just include the daily testing rate as a time varying covariate?

• Figures 3 and 4 would be improved with some sort of correlation coefficient in each panel. It is unclear if all of the panels in figure 4 were considered as candidate covariates. Why aren’t business restriction, stay at home mandates, and number of state actions in the final model? Were they not retained in stepwise? Were they too highly correlated?

Specific points

Abstract

• The abstract states “we examine how commonly used epidemiological metrics differ for each individual state within the US during the initial COVID-19 outbreak. Yet, the manuscript does not report out individual state level results so this sentence should be revised.

Methods

• “The early doubling time should not be severely affected by government interventions s these were rarely implemented that early in the course of the epidemic” (line 65-66) I think the sentiment of this statement is true… but it is likely more that the effects of the policies wouldn’t take place that fast. Several states closed schools and implemented gathering bans when they had very very few cases (e.g. West Virginia closed schools with 0 cases). Given that it would take 10+ days for policy change to impact coronavirus spread (see lag in model derived by Auger, JAMA), I believe this is a reasonable approach; however, you may want to tweak the language because it isn’t that the policies weren’t implemented it is more they wouldn’t have had time to take effect.

Results

• “We found that doubling times for all states increased with time and that heterogeneity between states was reduced” line 109-110. This seems very subjective. It is strange not to quantify this in some way. How much did they change?

Discussion

• The first paragraph of the discussion is a bit long and feels tangential to the main point of the manuscript. Perhaps this could be largely reduced and additional references regarding emerging COVID studies (listed above) could be incorporated. There are some nice additions (or considerations) in this manuscript which haven’t been in other models (tightness, volunteer rate, etc). So highlighting these is appropriate (as you do in lines 170-180).

Supplemental figure 3

This one is a bit confusing to me. You talk about the changes in the doubling time throughout the study period; yet, this is just a single number per state... so when is this from? Early? The average of the study period???

Conclusions

• appropriate

Reviewer #3: Summary

The reviewed paper presents an interesting analysis on how state-level variation can explain the early pandemic dynamics of COVID-19. The analysis appears to be logical and statistically sound. I don’t have any major comments, but have a few minor comments that I think could enhance the manuscript.

Major comments

N/A

Minor comments

The authors note: “We show that early non-pharmaceutical government actions were the most important determinant of epidemic dynamics.” I think this statement can be interpreted in two ways: (1) government actions need to happen early to impact the epidemic dynamics, or (2) the government actions that happened early had the largest impact on subsequent epidemic dynamics. These have two different interpretations in my mind, that the specific timing in the early phase matters or that implementing them early at all matters. I believe the authors mean the latter, but I think it would be useful to clarify, maybe something like this?: “We show that non-pharmaceutical government actions during the early phase of the epidemic were the most important determinant of epidemic dynamics.” One could reference previous research from China that agrees with this conclusion as well (I know it likely wasn’t published when this was written): https://wwwnc.cdc.gov/eid/article/26/9/20-1932_article

Figure 5: The authors note: “The overall doubling time was not strongly correlated with overall tests rates for each state (Fig. 4).”

Did the authors compare with testing rates as a raw number rather than per capita? I wonder if there would be a stronger correlation, as the early pandemic dynamics with small numbers might not have any per capita impact.

In general, I found it unintuitive to follow the doubling time results. E.g. on line 120: “We found that population density, flu vaccination rates, and wealth were all positively correlated with doubling times” I would suggest rephrasing for clarity for these types of results throughout to something like: “Comparing the doubling times across states, we found that increasing population density, flu vaccination rates, and wealth were all associated with slower epidemics” -- I’ll leave it to the authors to decide if they agree

Line 120 indicates that all three variables are positively correlated with doubling times, but the results in Table 1 have different signs for population density from the other two variables, so I think there might be a typo?

Line 173, I’d add that tightness was negatively correlated with doubling time.

Typo in line 180: “We hypothesis this may be the result of people in tight cultures finding it more difficult to adjust their behavior when new rules are imposed.”

7. PLOS authors have the option to publish the peer review history of their article (what does this mean?). If published, this will include your full peer review and any attached files.

Reviewer #2: No

Reviewer #3: No

---

## [Author Response · Author response to Decision Letter 1]

29 Sep 2020

We have included a revised manuscript and detailed response to reviewer concerns.

---

## [Editor Report · Decision Letter 2]

1 Oct 2020

State-level variation of initial COVID-19 dynamics in the United States

PONE-D-20-13138R2

Dear Dr. White,

We’re pleased to inform you that your manuscript has been judged scientifically suitable for publication and will be formally accepted for publication once it meets all outstanding technical requirements.

Kind regards,

Jeffrey Shaman

Academic Editor

PLOS ONE
---

## [Editor Report · Acceptance letter]

5 Oct 2020

PONE-D-20-13138R2 

State-level variation of initial COVID-19 dynamics in the United States 

Dear Dr. White:

I'm pleased to inform you that your manuscript has been deemed suitable for publication in PLOS ONE. Congratulations! Your manuscript is now with our production department. 

Kind regards, 

on behalf of

Prof. Jeffrey Shaman 

Academic Editor

PLOS ONE